# Characteristics of Stormwater Quality in Singapore Catchments in 9 Different Types of Land Use

**Haihong Song [1,*], Tingchao Qin [1], Jianbin Wang [2] and Tony H. F. Wong [2]**

[1]   Department of Civil and Environmental Engineering, College of Engineering, Shantou University, Shantou 515000, China; 16tcqin@stu.edu.cn

[2]   Cooperative Research Centre for Water Sensitive Cities, Level 1, 8 Scenic Blvd, Clayton Campus, Monash University, 3800 Melbourne, Australia; wang.jianbin@crcwsc.org.au (J.W.); tony.wong@crcwsc.org.au (T.H.F.W.)

[*]   Correspondence: hhsong@stu.edu.cn; Tel.: +86-754-8846-2466

**Abstract:** Stormwater quality is well known for its highly stochastic nature and not necessarily well explained by mechanistic urban build up and wash off models. Therefore, local empirical data (based on land use) are an essential compliment to statistical analyses of global data. This paper reports on a large-scale monitoring of the 12 key water quality parameters of suspended solids, nutrients, and heavy metals for stormwater runoff in urban discharges from nine urban land uses with varying sizes in Singapore. It was found that, in general, the average of the event mean concentrations for total nitrogen, total phosphorus, total organic carbon, total suspended solids (TSS), and phosphate in parkland land use were higher than the other eight studied land uses. Based on Pearson's correlation analysis, significant correlation between pairs of water quality parameters was observed. Particularly, there was significant correlation between TSS and most of the other tested water quality parameters in all land uses. A pollutant data set from this study will assist in developing appropriate stormwater quality models, guide the establishment of stormwater treatment objectives and preliminary designs for Singapore catchments, as well as provide an essential complement to statistical analyses of global data for stormwater characteristics.

**Keywords:** stormwater runoff; stormwater quality; event mean concentrations (EMCs); land use

## 1. Introduction

Urban stormwater conveys pollutants derived from natural and anthropogenic activities. It is a major source of surface water pollution in urban areas and one of the most important causes for the deterioration of water quality in the receiving water environment [1–4]. With urbanization, land-use development to support population increases and activities results in an increase of imperviousness to and a consistently declining health of the receiving water bodies [5–8]. It is therefore necessary to construct storm-water treatment systems to manage the risks associated with stormwater pollution [9,10].

Proper assessment of catchment runoff quality is essential for predicting pollution loads generated from urban areas to manage stormwater pollution. Previous studies show that pollutant concentrations and loads for individual watersheds depend on a few factors including land use [7,11–13], sewer system, type of surface drained, rainfall, and runoff [14]. Different land-use characteristics reflect anthropogenic activities and could have a major impact on the quality of stormwater and consequently the receiving water bodies. Stormwater quality is well known for its highly stochastic nature and local empirical data (based on land use) is essential for a more accurate prediction of pollutant load at a local level [15–17]. Therefore, specific local runoff water quality data is critically helpful not only for

the accurate runoff water quality model development but also for better understanding of the current and future impact of land uses change on downstream water bodies.

Many local studies have focused on establishing relationships between land use types and storm runoff pollution characteristics for various land uses and surface types including residential, industrial, commercial, highways, bridges, lawns, roads, roofs, and parking lots [18–25]. The composition of land use for a rapidly urbanizing catchment is usually heterogeneous, and this may result in significant spatial variations of storm runoff pollution [23]. It is therefore vital to address how various land uses impact pollutant loading, so that changes in water quality due to modified land uses can be more accurately predicted and realistically modelled, based on which proper stormwater pollution mitigation strategies can be strengthened [11,26].

Singapore is a tropical catchment with a high population density and water is a scarce resource [27]. The average annual rainfall in Singapore is between 1650 and 2550 mm with no dry season. It is, however, important to note that rainstorms over Singapore are usually local, brief, and intense. The humid tropics have experienced rapid urban growth which causes deterioration in waterway health [28]. In recent years, the Singapore government has placed increasing emphasis on developing and implementing strategies for better stormwater management [29,30]. In a Singapore catchment with pre-dominant urban residential land use, Lim et al. [28] observed a lower mean concentration of the suspended solids than the world data set [16]. Similarly, Chua et al. [14] studied the stormwater and dry weather flow water quality characteristics in four sub-catchments of different land uses in the Kranji catchments. In all the Kranji sub-catchments, with the exception of the predominantly rural land use sub-catchment, showed consistently lower total suspended solids, total phosphorus and total nitrogen concentrations than the global mean [16]. The locally-derived stormwater quality data suggested that Singapore catchments may have lower stormwater pollutant concentrations as compared with global data. This highlights the need for further region (Singapore)-based characterisation of stormwater quality data to validate this observation as it may have significant implications on the determination of appropriate stormwater quality treatment objectives.

The main objectives of this study are to undertake targeted catchment monitoring of stormwater pollutants to develop pollutant export statistics and relationships for various types of pollutants from a range of urban catchments (encompassing a range of land use types, storm events, and soil types). These statistical relationships define the stochastic nature of pollutant concentrations, including the cross-correlations between different pollutant constituents. Statistical analyses were undertaken to determine causal relationships between the land use and stormwater pollutant characteristics. This information will assist in developing appropriate pollutant data set relevant to Singapore catchments for urban stormwater quality models and will be invaluable in guiding the establishment of stormwater treatment objectives for Singapore catchments.

## 2. Materials and Methods

### 2.1. Catchments and Monitoring Program

A comprehensive catchment stormwater runoff monitoring programme was conducted at 9 sampling sites, where catchment area is relatively small with a distinct land use. The 9 sampling sites covered a range of typical land use types in Singapore including mixed use, residential, commercial (food centre and business district), industry (automobile and factory), road (major road and residential road), and parkland. The mixed land use type included a range of land uses such as commercial area, residential area, car park and road, typical of mixed land-use in Singapore (Figure 1).

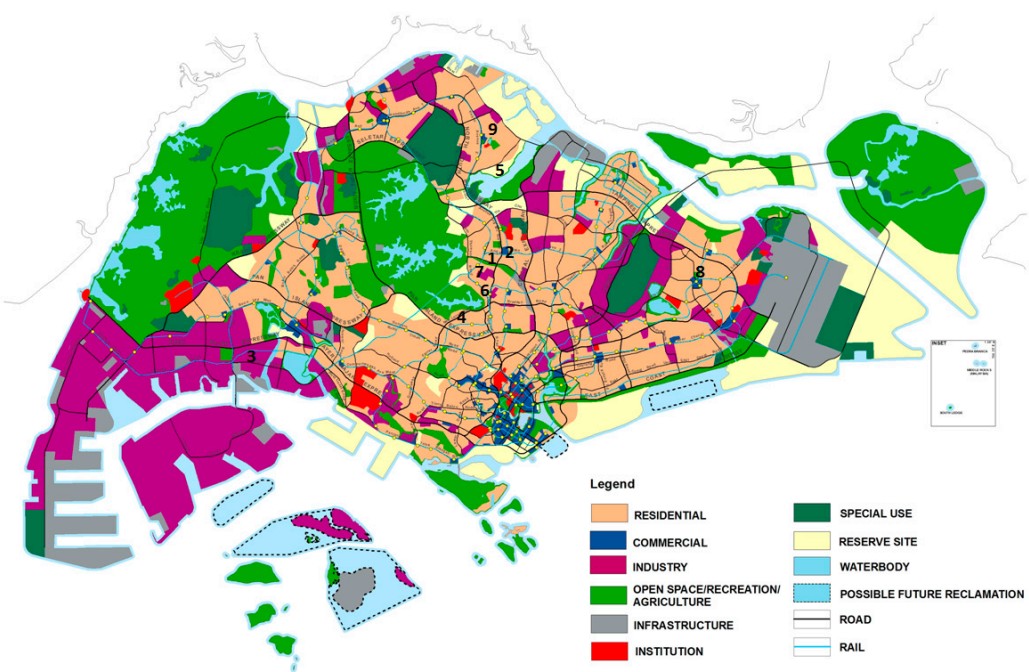

**Figure 1.** Sampling and monitoring sites: 1. AMK (mixed use, 4.50 ha); 2. ChengSan (commercial food centre, 0.37 ha); 3. ChinBee (industry, 5.70 ha); 4. Lornie (major road, 1.0 ha); 5. Lower Seletar (parkland, 1.0 ha); 6. Pemimpin (residential, 1.10 ha); 7. SinMing (car workshop, 2.10 ha); 8. Tampines (commercial business centre, 2.10 ha); 9. Yishun (residential road, 0.46 ha).

*2.2. Runoff Sample Collection and Analyses*

To evaluate water quality of the runoff, water sampling campaigns were planned to be conducted in the urban stormwater catchments during storm events from April 2012 to March 2013. However, due to many factors, first data collection started from August 2012. Monitoring stations were set up at the drainage channels of each sampling site. The stations were equipped with water level and velocity sensors (Sigma 950AV, Sigma-Hach Company, Loveland, CO, USA), rain gauges (Rimco8020, Campbell Scientific, Garbutt, Queensland, Australia), and auto-samplers (Isco3700, Teledyne ISCO, Lincoln, NE, USA). Data from the instruments were logged every 2 min using data loggers (CR800, Campbell Scientific, Logan, UT, USA). The auto-samplers were used to collect samples during a storm event and were triggered when the level in the storm drains reached a pre-set value. Afterwards, the samples were collected every 5 min by the auto-samplers in polypropylene sampling bottles (1 L). All water samples were brought back to laboratory for analysis within 24 h of collection. There were 8 sampling events for most of the sampling sites except the parkland and the food centre, which had 4 and 6 events respectively. For each event, 9 to 12 water samples were collected from the drainage channels of each sampling site during the storms. Chemical analyses for the water quality parameters were conducted in accordance with APHA Standard Methods for Examination of Water and Wastewater (21st Edition, 2005), and the methods stated by the USEPA.

*2.3. Calculation of Event Mean Concentrations (EMCs)*

Event mean concentrations (EMCs) of rainfall events were used to characterize pollutant concentration. It is defined as the total mass load of a pollutant from a site during a storm divided by the total runoff water column during the storm [21]. EMCs can be expressed as:

$$\mathrm{EMC} = \frac{\mathrm{M}}{\mathrm{V}} = \frac{\int_0^t \mathrm{C(t)Q(t)dt}}{\int_0^t \mathrm{Q(t)dt}} = \frac{\sum \mathrm{C(t)Q(t)dt}}{\sum \mathrm{Q(t)dt}} \tag{1}$$

where, M is the total mass of pollutant during the entire runoff event (kg), V is the runoff volume during the storm event (m$^3$), C (t) is time varying pollutant concentration (mg/L); Q (t) is time variable flow (L/s); and t is total duration of runoff (s). A velocity–water depth rating curve was established for each sampling site based on application of the Manning's formula calibrated to observed velocity and water depth values of all events. The rating curves were used to back fill the observation gaps in velocity to calculate flow and runoff volume needed for the calculation of EMCs.

*2.4. Statistical Analyses*

To compare the differences of the average of EMCs for each pollutant constitutes from different land uses, one-way analysis of variance (ANOVA) was conducted. To examine the association between all pairs of water quality parameters within each land use and for all land uses integrated, Pearson's correlation was analysed respectively. Statistics analyses were conducted using Sigmaplot 12.0 and SPSS (Statistical Package for the Social Sciences) 17.0.

## 3. Results

*3.1. Characteristics of the Runoff Water Quality*

The ranges for the average of EMCs for total nitrogen (TN), total phosphorus (TP), total suspended solids (TSS), and total organic carbon (TOC) from the 9 sampling sites was 0.93–3.02 mg/L, 0.05–0.31 mg/L, 17.23–147.34 mg/L, and 0.98–4.16 mg/L respectively (Table 1). TN and TP concentrations in all land uses except the parkland were lower than the global mean EMC reported by Duncan which was 2.80 mg/L and 0.36 mg/L respectively [16]. In parkland, the average of EMCs for TN and TP was 3.02 mg/L and 0.31 mg/L respectively. These were comparable to the global mean. Five of the 9 land uses showed the average of EMCs for TSS significantly below the global mean EMCs 153.5 mg/L [16]. The average for the other four land uses (mixed commercial area, residential area, parkland and residential road) are comparable to the global mean. The highest average of EMCs for TOC (4.16 mg/L) was observed at parkland. It was found to be reported below the global TOC range (13.27–43.75 mg/L) [16].

**Table 1.** The average of the event mean concentrations (EMCs) for water quality parameters.

| Land Use | Mixed | Resid-ential | Food Centre | Business District | Car Workshop | Industry | Resid-ential Road | Major Road | Parkland |
|---|---|---|---|---|---|---|---|---|---|
| TN | 1.85 (0.13) | 1.16 (0.21) | 0.93 (0.20) | 1.08 (0.15) | 1.56 (0.35) | 1.61 (0.23) | 1.11 (0.15) | 1.14 (0.19) | 3.02 (0.37) |
| TP | 0.17 (0.02) | 0.07 (0.01) | 0.08 (0.02) | 0.09 (0.02) | 0.07 (0.02) | 0.12 (0.02) | 0.13 (0.01) | 0.05 (0.01) | 0.31 (0.03) |
| PO$_4$-P | 0.03 (0.001) | 0.02 (0.001) | 0.02 (0.01) | 0.02 (0.001) | 0.02 (0.01) | 0.02 (0.01) | 0.02 (0.01) | 0.01 (0.001) | 0.14 (0.01) |
| NO$_3$-N | 0.72 (0.06) | 0.59 (0.17) | 0.38 (0.12) | 0.58 (0.13) | 0.78 (0.18) | 0.61 (0.19) | 0.29 (0.08) | 0.52 (0.11) | 0.75 (0.10) |
| NH$_4$-N | 0.27 (0.07) | 0.10 (0.02) | 0.19 (0.04) | 0.07 (0.01) | 0.14 (0.12) | 0.42 (0.12) | 0.11 (0.01) | 0.15 (0.03) | 0.24 (0.04) |
| TSS | 112.07 (13.70) | 31.92 (4.66) | 17.23 (4.70) | 50.84 (6.68) | 37.67 (11.89) | 35.57 (5.53) | 142.76 (32.68) | 23.84 (4.37) | 147.34 (35.30) |
| TOC | 2.04 (0.31) | 0.98 (0.11) | 1.08 (0.14) | 1.50 (0.36) | 1.97 (0.32) | 2.03 (0.45) | 2.33 (0.65) | 2.26 (0.24) | 4.16 (0.18) |
| Zn | 0.20 (0.03) | 0.06 (0.01) | 0.20 (0.05) | 0.45 (0.07) | 0.23 (0.05) | 0.38 (0.07) | 0.21 (0.06) | 0.13 (0.03) | 0.18 (0.08) |
| Cu | 0.02 (0.00) | 0.01 (0.00) | 0.01 (0.00) | 0.03 (0.01) | 0.02 (0.001) | 0.09 (0.03) | 0.02 (0.01) | 0.02 (0.001) | 0.12 (0.05) |
| Fe | 1.88 (0.32) | 1.43 (0.30) | 0.41 (0.07) | 1.24 (0.18) | 0.94 (0.24) | 1.89 (0.36) | 3.00 (1.13) | 0.68 (0.19) | 0.65 (0.34) |
| Mn | 0.05 (0.01) | 0.02 (0.001) | 0.02 (0.001) | 0.04 (0.01) | 0.02 (0.001) | 0.04 (0.01) | 0.04 (0.01) | 0.02 (0.001) | 0.02 (0.01) |
| Ni | 0.004 (0.004) | 0.003 (0.0001) | 0.003 (0.0001) | 0.004 (0.0003) | 0.004 (0.0006) | 0.01 (0.002) | 0.004 (0.0004) | 0.007 (0.002) | 0.005 (0.0007) |
| n | 8 | 8 | 6 | 8 | 8 | 8 | 8 | 8 | 4 |
| s | 89 | 86 | 61 | 90 | 75 | 86 | 89 | 80 | 39 |

Note: Number in brackets means the standard error, "n" is the number of sampling events and "s" is the number of total samples at each sampling site. TN: total nitrogen, TP: total phosphorus, TSS: total suspended solids, TOC: total organic carbon.

The range for the average of EMCs for $NO_3$-N, $NH_3$-N and $PO_4$-P was 0.3–0.8 mg/L, 0.45–0.8mg/L and 0.01–0.14 mg/L respectively. The ranges for the average of EMCs for the tested metals which are Zn, Cu, Fe, Mn and Ni was 0.06–0.45 mg/L, 0.007–0.12 mg/L, 0.41–3.00 mg/L, 0.02–0.05 mg/L, and 0.003–0.010 mg/L respectively (Table 1). The average EMCs for Zn and Cu at the industrial areas was higher than the other land uses. Also, the average pH at the industry car workshop area was 7.24 which was significantly higher than that in food centre, major road and residential area.

The highest average EMCs for $NO_3$-N (0.75 mg/L) was observed at the parkland but it was not significantly different from the other land uses, while the highest $NH_4$-N (0.42 mg/L) occurred in the industry area. The ratio of the average of EMCs for $PO_4$-P to TP in different land uses ranged from 16.5% to 48.7%, with the highest occurred in the parkland. The ratio of the average EMCs for $NO_3$-N and $NH_4$-N to TN was found to be 24.5–50.7% and 7.5–22.3%, respectively.

*3.2. Correlation between Water Quality Parameters*

3.2.1. Correlation between Water Quality Parameters within Each Land Use

TSS was positively and significantly correlated with most of the other water quality parameters within each land use. In particular, there was strong significant correlation between TSS and TOC in the parkland, residential road and the commercial business centre with r of 0.72, 0.77, and 0.89 respectively ($p < 0.0001$). TSS also had significant correlation with TP ($0.70 \le r \le 0.97$, $p < 0.0001$) in all land uses with a noticeable strong correlation ($r \ge 0.94$, $p < 0.0001$) in the commercial food centre, the parkland and the commercial business centre. Also, TSS was significantly correlated with TN ($0.53 \le r \le 0.87$, $p < 0.0001$) in all land uses except in the residential area and the industrial car workshop area. Noticeably, both TN and TP were highly correlated with TSS in the parkland and the commercial business centre with r > 0.84 ($p < 0.0001$).

TN and TP were positively and significantly correlated in all land uses except in residential area. In industrial car workshop area and major road, the correlation between TN and TP was weak but significant with r as 0.28 and 0.33 respectively ($p < 0.01$) while the correlation was significant and stronger in the other land uses ($0.58 \le r \le 0.87$, $p < 0.0001$).

TP and $PO_4$-P had significant correlation in commercial food centre and industrial area, with r as 0.56 and 0.72 respectively ($p < 0.0001$) while the correlation in mixed use area, commercial business centre and residential road was weaker but significant with r as 0.27, 0.35 and 0.34 respectively ($p < 0.0001$). There was no significant correlation between TP and $PO_4$-P in major road, parkland, or industrial car worksho

Generally, TN was positively and significantly correlated with nitrogen species $NO_3$-N and $NH_4$-N in all land uses except in parkland. In particular, TN was highly correlated with $NO_3$-N in commercial business centre, residential area, parkland and industrial car workshop area with r as 0.70, 0.86, 0.88, and 0.95 respectively ($p < 0.0001$). In the industrial area, TN was highly correlated with $NH_4$-N with r as 0.90 ($p < 0.0001$). However, in the parkland, TN was negatively correlated with $NO_3$-N (r = −0.50, $p < 0.01$) and positively correlated with $NH_4$-N (r = 0.88, $p < 0.0001$).

Additionally, there was high and significant correlation between all heavy metals (r close to or above 0.8, $p < 0.0001$) in all land uses. Also, there was significant correlation between conductivity and pH in all land uses (r close to or above 0.50, $p < 0.0001$) except in the commercial food centre and the industrial factory.

3.2.2. Correlation between Water Quality Parameters for All Land Uses Integrated

Overall, TOC was positively and significantly correlated with all the other water quality parameters except pH ($0.17 \le r \le 0.63$, $p < 0.0001$). TSS was significantly correlated with TN, TP, and TOC with r as 0.54, 0.73, and 0.56 respectively ($p < 0.0001$). The correlation between TSS and the heavy metals was positive and significant ($0.14 \le r \le 0.71$, $p < 0.0001$). However, while TSS was significantly correlated with $PO_4$-P, $NH_4$-N ($p < 0.001$), the correlation was weak with r as 0.24, 0.20. Interestingly, TN was

significantly correlated with all other water quality parameters except pH ($0.29 \leq r \leq 0.74$, $p < 0.0001$). In particular, TN had relatively high correlation with TP, $NH_4$-N and TOC with r as 0.74, 0.60, and 0.62 respectively ($p < 0.0001$). There was also positive and significant correlation between metal ions ($0.36 \leq r \leq 0.82$, $p < 0.0001$). In particular, Pb has high and significant correlation with other metals except Ni and Cd with r above 0.70 ($p < 0.0001$). A significant correlation between pH and conductivity (r = 0.54, $p < 0.0001$) was also observed (Table 2).

**Table 2.** Pearson's correlation coefficients (r) between the averages of event mean concentrations (EMCs) for water quality parameters when data of all land uses are integrated.

| | TN | TP | PO$_4$-P | NO$_3$-N | NH$_4$-N | TSS | TOC | Pb | Zn | Cu | Cr | Fe | Mn | Ni |
|---|---|---|---|---|---|---|---|---|---|---|---|---|---|---|
| pH | NS | **0.09** | NS | NS | NS | NS | NS | 0.11 | **0.12** | NS | NS | **0.12** | NS | NS |
| TN | | 0.74 | 0.42 | 0.45 | 0.60 | 0.54 | 0.62 | 0.29 | 0.52 | 0.31 | 0.29 | 0.41 | 0.53 | 0.34 |
| TP | | | 0.58 | NS | 0.54 | 0.73 | 0.63 | 0.45 | 0.59 | 0.43 | 0.39 | 0.56 | 0.67 | 0.30 |
| PO$_4$-P | | | | 0.12 | 0.21 | 0.24 | 0.28 | NS | NS | NS | **0.07** | NS | NS | NS |
| NO$_3$-N | | | | | | **−0.08** | 0.17 | NS | **0.13** | NS | NS | NS | NS | NS |
| NH$_4$-N | | | | | | 0.20 | 0.29 | 0.17 | 0.30 | 0.24 | 0.24 | 0.28 | 0.31 | 0.31 |
| TSS | | | | | | | 0.56 | 0.48 | 0.56 | 0.36 | 0.34 | 0.69 | 0.71 | 0.14 |
| TOC | | | | | | | | 0.57 | 0.59 | 0.54 | 0.46 | 0.49 | 0.68 | 0.31 |
| Pb | | | | | | | | | 0.81 | 0.82 | 0.76 | 0.71 | 0.74 | 0.36 |
| Zn | | | | | | | | | | 0.81 | 0.68 | 0.67 | 0.80 | 0.44 |
| Cu | | | | | | | | | | | 0.79 | 0.65 | 0.71 | 0.55 |
| Cr | | | | | | | | | | | | 0.61 | 0.64 | 0.54 |
| Fe | | | | | | | | | | | | | 0.78 | 0.36 |
| Mn | | | | | | | | | | | | | | 0.39 |

Note: Normal value means $p < 0.0001$ and value in bold means $p < 0.05$, which represents a significant correlation; NS means non-significant constant. The number of observations is 695.

## 4. Discussion

This study observed lower level of stormwater pollutants in Singapore catchments with different land uses compared to global data reported by Duncan [16]. This agrees with previous studies on the stormwater quality in Singapore by Lim [28] and Chua et al. [14]. Also, the stormwater pollutants level in Singapore catchments was lower than that from Malaysia, which has relatively similar climate and rainfall pattern. For catchments in Malaysia, Chow et al. [31] observed $NO_3$-N as 0.9 mg/L for residential area, 0.93 mg/L for commercial area, 1.2 mg/L for light industry area while Nazahiyah et al. [32] observed $NO_3$-N for residential area as 2.4 mg/L and 2.8 mg/L for commercial area, which were all considerably higher than that in the 9 catchments of the current study. However, the average of EMCs for $PO_4$-P in this study was 0.01–0.14 mg/L (Table 1), which was higher than that from Marina catchment area, Singapore at which $PO_4$-P concentration was 0.008 to 0.010 mg/L [33]. Lucke et al. [17] reviewed the latest publications on urban runoff pollutants from residential and commercial areas of South-East Queensland (SEQ), Australia. The results show stormwater pollutant concentrations in SEQ to be significantly lower than those historically published as typical for Australian land uses but still higher in nutrient levels than that of this study (Table 3).

The lower level of pollutants in Singapore catchments may be attributed to the effective pollutant source management system to minimize the pollution loads into the drainage system in Singapore. PUB (Public Utilities Board), being the national water agency, put in place integrated waste and wastewater management policies and anti-pollution measures to protect the quality of stormwater runoff. The measures adopted to minimize pollutants in the catchments include strict erosion control at construction sites, efficient street sweeping practices, routine drainage system maintenance, and separation surface runoff from the sewage [34]. By well-coordinated land use planning and integrated catchment management, major pollutant producing activities in the catchments were eliminated or controlled.

**Table 3.** Comparison of log-transformed EMCs for residential and commercial areas in South-East Queensland Australia [17] and Singapore.

| Land Use | Parameter | Recent studies in Australia | | | Parameter | Current Study | | |
|---|---|---|---|---|---|---|---|---|
| | | TSS $\log_{10}$ values (mg/L) | TP $\log_{10}$ values (mg/L) | TN $\log_{10}$ values (mg/L) | | TSS $\log_{10}$ values (mg/L) | TP $\log_{10}$ values (mg/L) | TN $\log_{10}$ values (mg/L) |
| Residential | Number of events | 220 | 220 | 220 | Number of events | 8 | 8 | 8 |
| | Mean | 1.47 | −0.87 | 0.04 | Mean | 1.48 | −1.17 | 0.04 |
| | Std. Dev. | 0.51 | 0.63 | 0.38 | Std. Dev. | 0.227 | 0.206 | 0.189 |
| Commercial | Number of events | 100 | 100 | 100 | Number of events | 16 | 16 | 16 |
| | Mean | 1.34 | −1.01 | 0.02 | Mean | 1.51 | −1.06 | −0.02 |
| | Std. Dev. | 0.429 | 0.58 | 0.106 | Std. Dev. | 0.375 | 0.195 | 0.217 |

Note: "Std. Dev." means standard deviation.

In this study, we found a significant difference in stormwater pollutant levels between different land uses. Generally, the parkland exhibited the highest average of EMCs for TSS TN, TP, TOC, and $PO_4$-P which were significantly higher than the other land uses (Table 1). Barley et al. [35] reported that there was no statistically significant difference ($p < 0.05$) between data collected at plot and catchment scales for the same land use based on collating published and unpublished runoff constituent concentration and load data for Australian catchments covering 13 different land uses. This together indicated that differences in land uses is an important factor responsible for changes in the runoff constituents. However, whether it is accurate to generalize the water quality data in one catchment to the other catchment of the same land use for predication of pollutant load or modelling should be considered carefully. This is because both pollutant build-up and wash-off processes can be influenced by a range of catchment characteristics including conventional factors like land use and impervious area fraction as well as site specific characteristics such as urban form and impervious area layout [36].

The Pearson's correlation between water quality parameters showed that TSS was significantly correlated with most of the other water quality parameters in all land uses. This corresponds to the study of Wijesiri et al. [37] which suggested that pollutant wash-off is a function of the build-up of particles. The variations in pollutant load and composition during build-up are primarily determined by the temporal variations in particle size fractions. Therefore, TSS might be the dominant contributor for the variation of water quality in different land uses in Singapore catchments.

Although we observed a higher level of TN, TP, and ratio of $PO_4$-P to TP (48.7%) in the parkland, it is not statistically reliable to conclude that nutrients level in the runoff of parkland is significantly higher than the other land uses, considering the low number of events sampled in parkland. However, it is worthy to note that heavy storms might cause soil erosion in parkland. During soil erosion processes, sediment might be trapped and the sediment-attached nitrogen and phosphorus could be dissolved and transported by overflow in heavier rainfall [23]. Nitrogen and phosphorus should not be neglected in runoff treatment as they are responsible for eutrophication in water bodies. The practical implication is that the runoff generated at the parkland cannot be bypassed to receiving water bodies but must be treated, especially for the nutrients associated with the suspended solid.

The observed higher average of EMCs for Zn, Cu, and pH at industrial areas indicated the special contamination sources in these land uses. The sources of heavy metals in the storm water could be from dust particles from combustion plants, iron and steel industry, non-ferrous metal industry, waste incineration plant, cement industry, glass industry, and vehicle traffic [38]. The practical implication is that treatment of the runoff generated at the industry area should have an emphasis on the removal of heavy metals.

This study observed significant and positive correlations between water quality parameters. The correlation within each land use was generally stronger than that from all land uses integrated. The significant correlation between pairs of water quality parameters indicated that each pair of

pollutant loads might have the same sources (or washed off from the same land-use patterns). The practical significance is that any one of the pairs of pollutants can be chosen as representative or control indicators in runoff pollution monitoring and management for the sake of simplifications. However, the non-significant correlation observed between certain pairs of water quality parameters in each land use may simply reflect an inadequate sample size but did not prove the absence of any underlying relationshiAlso, the stronger positive correlation between pairs of pollutants within each land use might be because the pollutants have similar source, build-up and wash-off processes while in the general catchments, the source of contamination might be more variable and therefore the correlation between the pairs of pollutants becomes weaker. Hence, to establish the correlation between water quality parameters for either water quality monitoring purpose or for runoff pollutants immigration, it is more accurate to investigate the correlation within each land use.

## 5. Conclusions

This study was undertaken to develop pollutant export statistics and relationships for various types of pollutants in stormwater from a range of urban catchments encompassing a range of land uses in Singapore. It was found that the event mean concentrations (EMCs) of key pollutants for various land uses in Singapore were lower than the global mean. In general, the average of EMCs for TN, TP, TOC, TSS, and $PO_4$-P in parkland were higher than the other land uses. Based on Pearson's correlation analysis, significant correlation between pairs of water quality parameters was observed. In particular, there was significant correlation between TSS and most of the other tested water quality parameters in all land uses. Therefore, efficient capture of sediment would remove considerable pollutants from Singapore stormwater runoff. The information developed from this study will assist in developing appropriate pollutant data set relevant to Singapore catchments for urban stormwater quality models and will be invaluable in guiding the establishment of stormwater treatment objectives for Singapore catchments. The stormwater pollutant data set will also be an essential compliment to statistical analyses of global data for stormwater characteristics. However, it should be noted that sampling in this study does not provide adequate coverage of the potential variability that may occur during an annual rainfall pattern due to limited sampling rainfall events. A monitoring scheme of longer-term and more sampling events will provide more accurate pollutant data set for adoption.

**Author Contributions:** Conceptualization, H.S. and T.H.F.W.; methodology, H.S.; formal analysis, H.S.; resources, J.W.; data curation, T.Q.; writing—H.S. and J.W.; writing—review and editing, T.H.F.W.; funding acquisition, T.H.F.W. and H.S.

**Funding:** This research was funded by Guangdong Science and Technology Department (grant number 140/14140101 and 925/38040223) and a faculty star-up funding for Haihong Song by Shantou University.

**Conflicts of Interest:** The authors declare no conflict of interest.

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
