# Peer review of "Characteristics of Stormwater Quality in Singapore Catchments in 9 Different Types of Land Use"

_water, doi:10.3390/w11051089_

Round 1

Reviewer 1 Report

The manuscript reports on a large-scale monitoring of key water quality parameters of suspended solids, nutrients and heavy metals for stormwater runoff in urban wet-weather runoff from different urban land uses in Singapore.

The article is an original contribution and the topic is of interest for the readership of the Water journal.

English language is clear and the presentation is good; anyway, I have detected some criticisms in the text that should be properly addressed. A more detailed description is required for the monitoring protocol and instrumentation.

The Authors can benefit from the comments below to improve their paper. These have to be accomplished before manuscript acceptance.

Abstract

The abstract is concise and reflects the content of the article.

Introduction

Aims of the study are properly clarified in the Introduction.

Lines 35-37: concerning the negative effects of land use development on hydrology and water quality, the Authors are recommended to consider the following study as part of the introductory discussion:

-       Todeschini, S. (2016). Hydrologic and Environmental Impacts of Imperviousness in an Industrial Catchment of Northern Italy. Journal of Hydrologic Engineering, 21(7), 05016013, doi:10.1061/(ASCE)HE.1943-5584.0001348.

Lines 37-38: concerning the necessity to manage the risks associated with stormwater pollution, the Authors are recommended to consider the following study as part of the introductory discussion:

-       Todeschini S., Papiri S., Ciaponi C. (2018) Placement strategies and cumulative effects of wet-weather control practices for intermunicipal sewerage systems. Water Resources Management, 32(8), 2885-2900, doi: 10.1007/s11269-018-1964-y.

Line 59: A range for the annual rainfall over Singapore is indicated. Please specify the meaning of the provided minimum and maximum value.

Materials and Methods

This section is clear but not adequately detailed. Subsection 2.2. should be revised in order to clarify how storm events were selected and how the monitoring protocol was established (start and end of monitoring, sampling rate, …). Also the monitoring equipment should be briefly described.

Results and Analysis

Results are presented in a logical sequence. The provided tables are necessary for the understanding of the results.

Line 125: the meaning of “global EMC” is not clear. Please, replace “global mean EMC” with “global mean EMC reported by Duncan [14]”.

Line 154: Replace “TN and TP was” with “TN and TP were”.

Line 173: I suggest to quote Table 2 is subsection 3.1.2. Some of the r values indicated in the text do not correspond to the values provided in Table 2 (e.g., for TOC Table 2 shows 0.17<r<0.63 not 0.31<r<0.68). Please, check and correct.

Line 186: the variable “n” is not present in Table 1. Replace “n is the number of …” with “the number of …”.

Line 199: Replace “uses integrated” with “uses are integrated”.

Line 200: Replace “which represent” with “which represents”.

Line 266: “each land use might”  … maybe something is missing. Please, check the sentence.

Conclusions

Conclusions seem reasonable and are supported by the results.

References

Two references are suggested in the “Introduction” Section on the negative impacts of land use development and required wet-weather control. Apart from these references, based on my knowledge, no important reference is missing.

Author Response

Thank you very much for your comments for our manuscript. Please see the attachment.

Reviewer 2 Report

The manuscript is an interesting discussion and evaluation of the stormwater quality from several Singapore sub-catchments. It confirms findings that are being published in other areas internationally and is worthy of publication subject to some modification.

A few comments and suggestions follow;

Line 46 - this sentence is confusing. Suggest it be re-phrased;

Line 47 - the phrase "....data is be critically helpful...." is unclear. Suggest it be re-phrased;

Line 79 - are the relationship "casual" or "causal"?

Line 93 - it is unclear what the authors mean by a "flatted factory". Can this be described differently?

Section 2.2 - The manuscript does not detail the sample collection methodology. Several studies have found that the sampling methodology has been a significant influence on the EMCs observed, and subsequently the ability to compare with previous research. Suggest the manuscript could be improved by expanding on the sampling methodology (eg. grab samples/automated sampling);

Section 2.2 & 4 - the manuscript identifies that the parkland EMCs are different in composition to the other landuse types, but only 4 events were collected. Do the authors consider that 4 events are statistically robust enough to include in the data evaluation? Perhaps the parkland and food centre data should be evaluated more circumspectly given the low number of sampled events.

Section 2.2 - Sampling was undertaken from August 2012 to March 2013 (8 months). Do the authors consider that sampling for less than one full year provides adequate coverage of the potential variability that may occur during an annual rainfall pattern? It is suggested that this should be expanded in the manuscript.;

Line 166 - refer earlier point about "flatted";

Table 1 - mentions n in the footnote and indicates that it is between 4 and 8. Suggest that the table would be improved by adding a row for n and identifying n for each catchment;

Section 4 - The discussion makes reference to several documents that are more than a decade old and yet does not refer to recently published papers such as Lucke et al (2018) that present findings of comparable, recent research from Australia. Suggest a table comparing the findings of the Singapore study against this reference would be valuable;

Line 224 - it seems the Bartley Speirs reference is incorrectly numbered;

Lines 241 & 242 - Should these sentences be qualified given the low number of events sampled?

Lines 246 and 248 - This sentence seems out of place given the manuscript does not present the data in DIN and DIP terms. And it seems to be contradicted by the subsequent sentence that suggests the nutrients associated with the suspended solids are more important;

Section 5 - The Conclusion indicates that the observed EMCs were lower than the global mean reported from research more than 20 years old. Suggest the paper would be more valuable by referring to more recent published research;

Section 5 - The manuscript seems to have missed an important conclusion from the correlation results, namely that efficient capture of sediment would appear to also remove considerable nutrients in the Singapore case;

Lines 279 - 281 - Do the authors consider that information from 4-8 events on these catchments is sufficiently statistically robust to develop appropriate pollutant datasets  for guiding the establishment of stormwater quality models?

I look forward to the revised manuscript.

Round 2

Reviewer 1 Report

The manuscript has been significantly improved following the recommendations of the Reviewers; all my concerns have been addressed and convincingly justified.

Author Response

Thank you for your appreciation of our work.

Reviewer 2 Report

The manuscript has improved. There are still a few small amendments that, subject to correction, should not prevent publication;

Line 44 and 45 - the sentences appear to repeat.

Line 100 and 101 - is the sentence "...with water level and velocity and sensors...." correct, or should it read "...with water level and velocity sensors..."?

Line 262 - the sentence is unclear "....and it is therefore should not be neglected...."

Line 281 - should the sentence read "....each land use might be because the pollutants....."

Author Response

Thank you very much for your valuable suggestions for our manuscript. Please see the attachment.
